# Enhanced Screen Content Image Compression: A Synergistic Approach for Structural Fidelity and Text Integrity Preservation

## ABSTRACT

With the rapid development of video conferencing and online education applications, screen content image (SCI) compression has become increasingly crucial. Recently, deep learning techniques have made significant progress in compressing natural images, surpassing the performance of traditional standards like versatile video coding. However, directly applying these methods to SCIs is challenging due to the unique characteristics of SCIs. In this paper, we propose a synergistic approach to preserve structural fidelity and text integrity for SCIs. Firstly, external prior guidance is proposed to enhance structural fidelity and text integrity by providing global spatial attention. Then, a structural enhancement module is proposed to improve the preservation of structural information by enhanced spatial feature transform. Finally, the loss function is optimized for better compression efficiency in text regions by weighted mean square error. Experimental results show that the proposed method achieves 13.3% BD-Rate saving compared to the baseline window attention convolutional neural networks (WACNN) on the JPEGAI, SIQAD, SCID, and MLSCID datasets on average. Our code will be available after the paper is accepted.

## CCS CONCEPTS

• **Information systems**;

## KEYWORDS

Image compression, screen content image, external prior, structural fidelity, text integrity

**ACM Reference Format:**
. 2018. Enhanced Screen Content Image Compression: A Synergistic Approach for Structural Fidelity and Text Integrity Preservation. In *Proceedings of Make sure to enter the correct conference title from your rights confirmation email (Conference acronym 'XX)*. ACM, New York, NY, USA, 8 pages. https://doi.org/XXXXXXX.XXXXXXX

## 1 INTRODUCTION

With the rapid development of information and communication technology, image compression plays a key role in various applications such as video conferencing (e.g., Zoom, Webex), online education (e.g., Google Classroom, Kahoot), social media (e.g., WeChat, Facebook), etc. It can store or transmit images efficiently with limited hardware resources. Classical image compression methods follow a scheme including prediction, transform, quantization, and entropy coding [28]. The representative standards are JPEG [34], JPEG2000 [12], intra frame coding of H.264/HEVC/VVC [7, 31, 38]. These standards are handcraft designed, which no longer meet the increasing requirements of image compression. Motivated by this, learned-based image compression models are proposed and show better rate-distortion performance than conventional methods on metrics of both peak signal-to-noise ratio (PSNR) and multi-scale-structural similarity index (MS-SSIM) for natural images [42]. However, little attention has been paid to screen content image (SCI) compression, where the SCIs are spread everywhere across the Internet.

SCIs typically comprise text, graphics, and camera-captured natural images. Computer-generated text and graphics account for more than 90% of SCIs. In contrast to natural images, SCIs are characterized by their sharp edges, limited color palette, high contrast, noise-free, and significantly different region complexity [26]. Because of these distinctive attributes, conventional compression algorithms designed primarily for natural images are less effective when applied to SCIs. A significant loss of high-frequency information occurs during traditional compression, which seriously damages the quality of the reconstructed screen content due to edge and text blurring. As the human eye is sensitive to edge and text information, maintaining the high-quality compression of these details is crucial.

Recently, numerous schemes have emerged to improve the coding efficiency of SCIs. In the latest generation of coding standards, such as VVC, AV1 [9], and AVS3 [39], numerous technologies have been designed specifically for SCI. For example, VVC adopts intra-block copy (IBC) to handle regions with regular patterns and repetitions in SCIs. The use of palette mode to depict a coding block is based on the simplicity of color schemes in SCIs, contrasting with the complex and diverse colors found in natural images. Furthermore, VVC employs transform skip, adaptive color transform, and block-based differential pulse code modulation to adapt to the characteristics of SCIs [27]. Through these optimizations, VVC achieves a notable 33.22% Bjøntegaard-Delta Rate (BD-Rate) [6] reduction when compressing SCIs [26]. Currently, deep learning-based image compression methods show excellent performance. However, it is essential to highlight that existing end-to-end image compression solutions are designed for natural images, with limited attention paid to SCI compression. Besides, SCIs show different characteristics compared to natural images. Therefore, algorithms for natural images do not perform optimally on SCIs. To resolve this problem, some researchers proposed learned SCI compression. Wang et al. [35] propose a transform skip-inspired end-to-end SCI compression scheme. However, this scheme does not exhibit performance superiority when compared to VVC. Heris et al. [19] propose a segmentation network-based compression scheme. However, the framework does not consider the SCI characteristics when designing the network architecture.

In this paper, we propose a synergistic learned compression approach for SCI. This work aims to address the following three challenges: 1) resolving the global spatial structural information loss which is not considered in the window attention module (WAM), 2) resolving the improper bit allocation for SCI compression, 3) minimizing text region blurring in the reconstructed image. Experimental results show that the proposed method achieves 13.3% BD-Rate saving compared to the baseline window attention convolutional neural networks (WACNN) on the JPEGAI, SIQAD, SCID, and MLSCID datasets on average. The main contributions of our work are summarized as follows:

(1) The external prior guidance (EPG) module is proposed to capture global spatial attention. The basic structural and text position priors are combined to distinguish text regions from the rest, which helps to preserve structural fidelity and text integrity.

(2) A structural enhancement module (SEM) includes the feature extraction module (FEM) and nonlinear enhancement spatial feature transform (NESFT) is proposed to improve the preservation of structural information by enhanced spatial feature transform. The FEM can distinguish spatial structures intricately embedded within external priors and NESFT is proposed to propagate the structural prior features into the backbone.

(3) In the loss function, mean square error (MSE) is replaced by a weighted mean square error (WMSE) based on the text region mask. The proposed overall framework shows significant BD-Rate reduction over existing methods.

The rest of this paper is organized as follows. Section 2 provides a review of related works on end-to-end image compression, conditional compression, and SCI compression. Section 3 presents the proposed synergistic approach for structural fidelity and text integrity preservation. The experiment results are shown in Section 4, followed by the conclusion in Section 5.

## 2 RELATED WORKS

This section first briefly reviews several popular learned image compression algorithms. Then, we describe the conditional compression, which is closely related to the method in our proposed algorithm. Finally, we talk about SCI compression, which is the research focus of our study.

**Learned image compression:** In recent years, there has been remarkable progress in the field of learned image compression. Ballé et al. [2] first propose an end-to-end optimized model for image compression. Then, Ballé et al. [3] introduce a hyperprior architecture to effectively capture the spatial dependencies in latent features, thereby reducing spatial redundancy. With the success of auto-regressive priors in probabilistic generative models, Minnen et al. [24] introduce an auto-regressive module to improve coding efficiency. As the entropy model has great potential in optimizing image compression performance, numerous studies are working toward improving the module of entropy models [10, 14, 17, 18, 25]. Cheng et al. [10] employ discretized Gaussian mixture likelihoods, and Fu et al. [14] propose a Gaussian-Laplacian-logistic mixture model. In addition, there are some works to adjust network architecture. Liu et al. [22] propose a Non-Local Attention Module (NLAM)

to capture both local and global correlations. Contrarily, Zou et al. [42] propose the WAM to further capture correlations among spatial neighboring features.

**Conditional compression:** Conditional compression is typically employed in variable bitrate encoding, which aims to compress images at different bitrates using a single model [20]. Conditional compression is often achieved by altering the distribution of features. Cui et al. [11] introduce gain units to scale the latent representation in the channel dimension, thereby enabling adaptive continuous bitrate. Song et al. [30] utilize spatial feature transform (SFT) layers [37] to apply element-wise affine transform to intermediate features, enabling variable bitrate control and Region of Interest (ROI) encoding. Wang et al. [36] implement channel-wise scaling across all self-attention layers to achieve variable bitrate. It is worth noting that, as far as our knowledge, the previous works in this domain primarily aim at achieving variable bitrate, with no explicit exploration into the compression performance of the model. In our work, we alter the distribution of features with the external prior guidance to enhance learned SCI compression for better rate-distortion performance.

**Screen content image compression:** The video coding standards, HEVC and VVC, have introduced techniques such as the palette mode, cross-component prediction, and transform skip to improve the compression efficiency of SCI. In existing palette coding approaches, separate palette coding is applied to the luma and chroma components. Then, Vishwanath et al. [33] consider the correlation between different components and propose cross-component palette prediction using luma for chroma. Ramin et al. [15] propose the Joint Cross-Component Linear Model (J-CCLM) to improve prediction efficiency by combining the conventional CCLM mode with an angular mode derived from the co-located luma block. Tang et al. [32] combine text detection with HEVC-SCC to propose a text semantic-aware encoding strategy, achieving high text semantic fidelity at low bitrate. Inspired by the transform skip technique in traditional coding, an end-to-end transform skip image compression framework is introduced by Wang et al. [35], surpassing the SCI compression performance of the hyper-prior model in [3]. Heris et al. [19] introduce a segmentation network-based compression scheme to distinguish natural from synthetic content in SCI. Shen et al. [29] propose a pioneering entropy-efficient transfer learning module named Dec-Adapter to bridge natural image and SCI compression. However, the exploration of end-to-end SCI compression is limited, and its performance has not yet reached a level close to traditional methods. In our study, we will propose a structural fidelity and text integrity preservation SCI compression method to achieve superior performance compared to VVC.

## 3 METHODS

This section introduces our main idea of SCI compression for structural fidelity and text integrity preservation. Firstly, we describe the overall framework. Then, an EPG module is introduced. Finally, we discuss the SEM and the optimization of the loss function.

### 3.1 The overall framework

Our proposed network architecture is shown in Fig. 1, which is built based on the WACNN model [42]. The EPG is introduced to

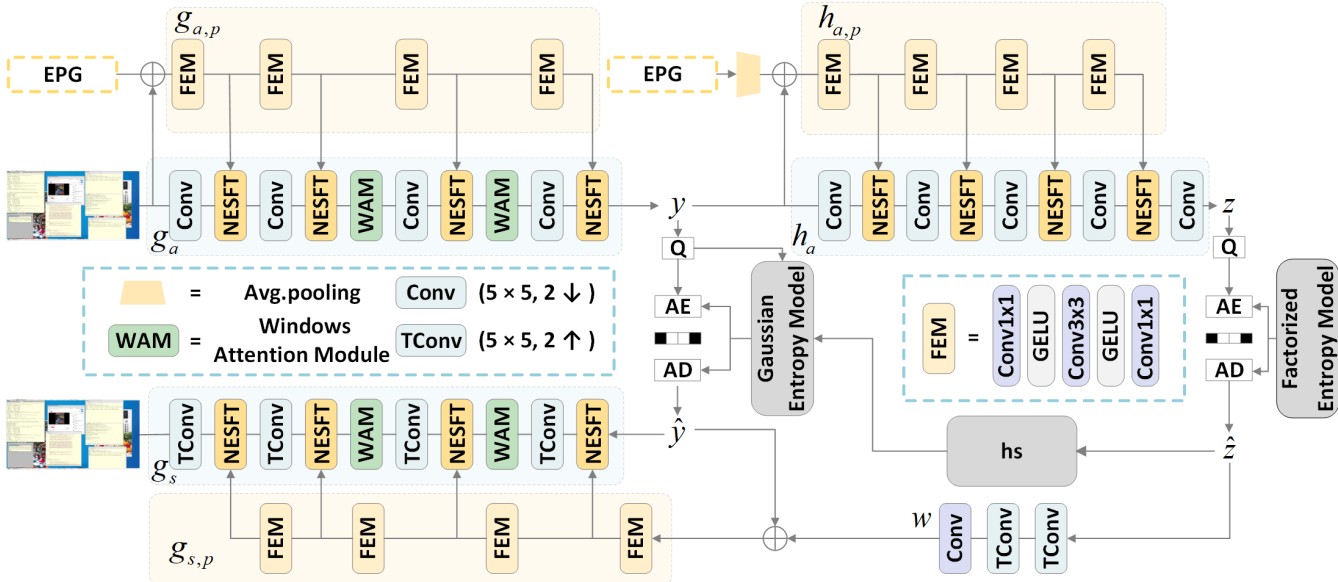

**Figure 1: The network architecture of the proposed framework.**

provide global spatial attention which is not considered in WACNN. The FEM is used to extract structural prior information of varying depths. The NESFT module is proposed to fuse the features of the FEM and the backbone convolutional layers by spatial affine transform. This is crucial for preserving the structural information of SCIs. The details of the network are described below.

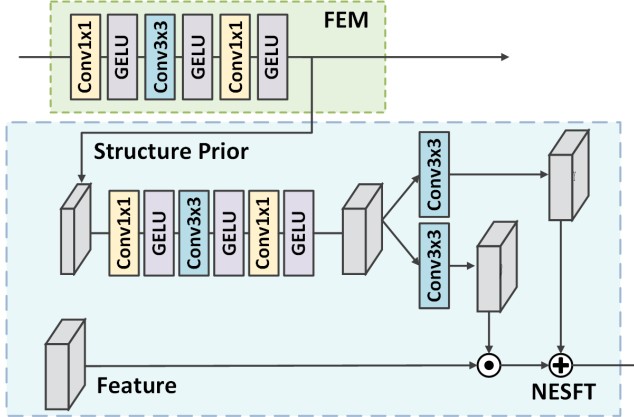

**Figure 2: Illustration of SEM. The parameters of element-wise scaling and shifting are generated from FEM, and then affine transform is performed.**

The main encoder $g_a$, built upon existing analysis transform modules, accepts the structural prior features that come from the prior branch $g_{a,p}$ and applies the spatial affine transform to intermedia features through the NESFT module. The main encoder $g_a$ and the prior branch $g_{a,p}$ take the image $x$ and the external prior $g'$ as inputs to generate the latent feature $y$, as formulated below:

$$y = g_a(x, \Phi_1), \qquad (1)$$

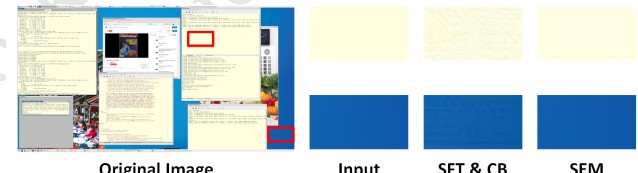

**Figure 3: Comparison of image reconstruction with the SFT & CB [30] and SEM.**

where $\Phi_1 = g_{a,p}(x, g')$.

The straight-through estimator (STE) strategy [5] is utilized to quantize the latent feature $y$, generating $\hat{y}$ for subsequent entropy coding. The formula is as follows:

$$\hat{y} = Q(y). \qquad (2)$$

The hyper-encoder also performs structural prior guided spatial affine transform, as shown below:

$$z = h_a(y, \Phi_2), \qquad (3)$$

where $\Phi_2 = h_{a,p}(y, g_1')$, with $g_1'$ being the down-sampling of $g'$ to ensure dimensional compatibility with $y$.

The entropy model is adopted the same as WACNN. The parameters $(\mu, \sigma)$ of the Gaussian entropy model are obtained from the hyperprior $\hat{z}$ to approximate the distribution of $\hat{y}$. The hyperprior $\hat{z}$ retains spatial importance information. Thus, our framework does not explicitly transmit structural prior, with consideration for bit saving. It recovers an approximate structural prior $w$ from the hyper prior $\hat{z}$. The main decoder $g_s$ takes the approximate structural prior $w$ and $\hat{y}$ as inputs to reconstruct the original image, as shown in the following formula,

$$\hat{x} = g_s(\hat{y}, \Phi_3), \qquad (4)$$

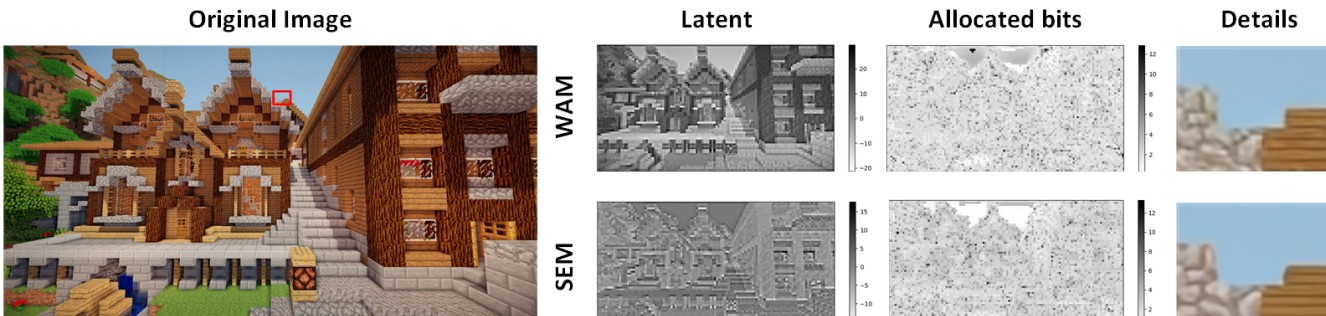

**Figure 4: Visualization of WAM and SEM for the channel with maximal entropy. It shows that our SEM focuses on the structure and allocates fewer bits on flat regions (such as the sky in computer-generated images). Instead, more bits are allocated to details than structure in the WAM.**

where $\Phi_3 = g_{s,p}(\hat{y}, w)$.

## 3.2 External Prior Guidance

SCIs exhibit different characteristics from natural images, where simple regions may appear completely flat and complexity varies significantly between different regions. When applied to SCIs, the WAM struggles to effectively capture and process global structures, as it focuses more on details within the window. To address this challenge, we propose EPG to optimize image compression by utilizing global attention. The external priors are highly correlated with the characteristics of SCIs, ensuring the preservation of structural fidelity and text integrity. These external priors consist of basic structural prior and text position prior.

Firstly, the structural information obtained by the Laplacian gradient operator is introduced as a basic structural prior. It can enhance the network's ability to preserve image edges and details, especially sharp edges and noticeable structural transition areas commonly found in SCIs. The Laplace gradient operator is defined as follows,

$$\nabla = \begin{bmatrix} 1 & 1 & 1 \\ 1 & -8 & 1 \\ 1 & 1 & 1 \end{bmatrix}. \tag{5}$$

We do not consider gradient direction information, as gradient magnitude is sufficient to reveal image structural information.

Then, text position prior is introduced to refine the basic structural prior, aiming at preserving text integrity. Using the text detection method, we detect text regions and translate them into binary masks, with text regions denoted as 1 and non-text regions donated as 0. Employing this text mask, the structural prior of images is introduced to enhance their representation of text regions. The formula is shown as follows:

$$g' = \frac{(M + \epsilon) \times g}{\epsilon + 1}, \tag{6}$$

where $g$ represents the basic structural prior, and $\epsilon$ is a predefined constant for fine-tuning the attention toward text regions. The revised priors enhance the contrast between text and non-text regions, aiding the network in distinguishing text regions from the rest.

To generate the text region mask, we evaluate three representative text detectors, including EAST [40], CRAFT [1], and RDD [21]. By the evaluation results in section 4, CRAFT is chosen as our text detector.

## 3.3 Structural Enhancement Module

To preserve the structural information of the image, we design a SEM. Altering the distribution of features is widely utilized in conditional compression for variable bitrate coding. However, the SEM is proposed to achieve a high coding performance instead of achieving variable bitrate coding. This module integrates both FEM and NESFT modules, as shown in Fig. 2. We craft the network architecture to endow the FEM with efficient feature extraction capabilities, enabling it to distinguish spatial structures intricately embedded within external priors. Leveraging a strategic combination of 1×1 convolution, we can enrich the network's nonlinear capabilities, significantly enhancing its capacity to handle complex external priors and affine transform relationships. We utilize NESFT to propagate the structural prior features extracted by the FEM into the backbone, instead of directly concatenating them with RGB features. This strategy ensures the efficient utilization of structural priors, augmenting feature representation in complex image regions and optimizing compression efficiency of flat regions. NESFT dynamically generates parameter pairs $(\gamma, \beta)$ based on the output of FEM, subsequently employing them to execute spatial affine transform on the features. The feature transform formula is shown as follows:

$$NESFT(F, \Phi) = \gamma \times F + \beta, \tag{7}$$

where F represents the intermediate features. As shown in Fig. 3, the spatial feature transform (SFT) and conditional block (CB) proposed by Song et al. [30] fails to effectively extract spatial structural information from structural priors, resulting in noticeable stripe artifacts in reconstructed images. We visualize the channel with the highest entropy for WAM and our method in Fig. 4. In the third column, it's obvious that our method can allocate fewer bits in simple regions and more bits in complex regions. However, WAM is constrained by the window-based mechanism, which allocates bits to flat regions. Moreover, the reconstructed images from our approach show clearer structure and fewer artifacts.

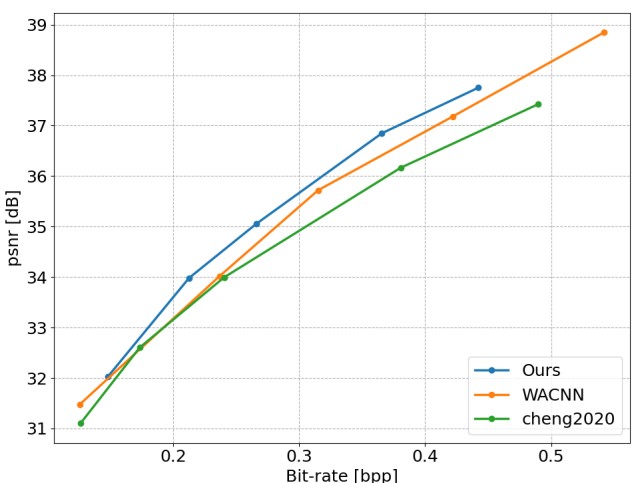

Figure 5: RD Performance evaluation on JPEGAI validation dataset.

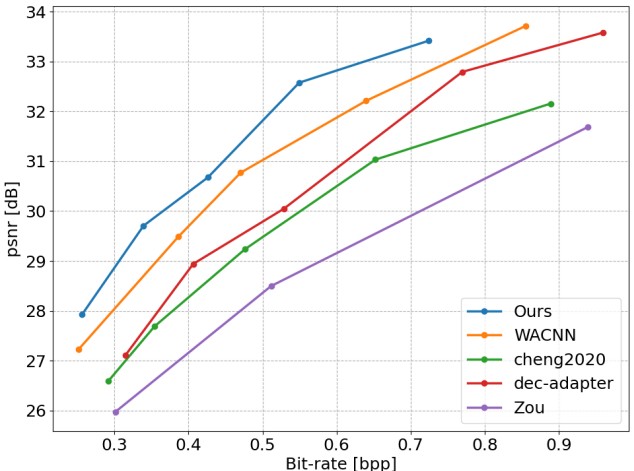

Figure 6: RD Performance evaluation on SIQAD dataset, which contains 22 high-resolution and high-quality images.

## 3.4 Rate-distortion Loss Function

To further enhance the compression efficiency of the network for SCIs, we make optimization to the loss function. The loss function for lossy image compression aims to achieve a balance between rate and distortion, which is defined as follows,

$$\mathcal{L} = \lambda \cdot D + R, \tag{8}$$

where $R$ represents the bitrate, $D$ represents the distortion, and $\lambda$ is a Lagrange multiplier that determines the trade-off between the bitrate and distortion. For text-priority compression demands, we introduce a refined loss function as follows,

$$\mathcal{L} = \Lambda^T \cdot D + R. \tag{9}$$

where $\Lambda = (\lambda_i)_{(i=1:N)}$ is the Lagrange multiplier matrix. It is no longer a single constant but a set of Lagrange multipliers dynamically adjusted by a text region mask $M = (M_i)_{(i=1:N)}$. Each element $\lambda_i$ is determined by a corresponding $M_i$ through a function mapping as follows,

$$\Lambda = (M/5 + 0.9) \times \lambda. \tag{10}$$

For the larger $M_i$, the mapping aims to assign greater $\lambda_i$ for the corresponding pixel $x_i$. Our designed loss function enables explicit spatial bit allocation in image compression based on $M$, giving priority to the compression quality of text regions.

The estimation of rate $R$ requires an entropy model $P$ conditioned on $M$. The entropy model outputs the likelihood of latent representation $\hat{y}$. This allows for dynamic adjustments based on the local features of the image, achieving targeted compression for text regions instead of applying a single strategy to the entire image. The MS-SSIM used as the distortion item in image compression will cause poor text reconstructions [23]. Thus, weighted mean squared error based on the text region mask is chosen as our metric for distortion evaluation. The formula is shown as follows,

$$\mathcal{L} = -logP(\hat{y} \mid M) + \sum_{i=1}^{N} \lambda_i \frac{(x_i - x_i')^2}{N} \tag{11}$$

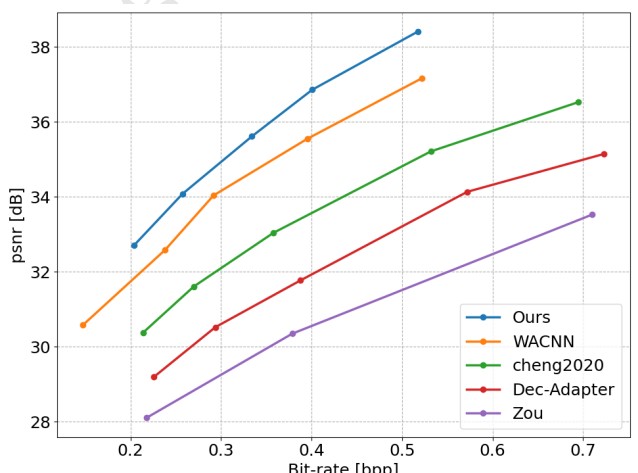

Figure 7: RD Performance evaluation on SCID dataset, which contains 200 high-resolution and high-quality images.

where $\hat{y}$ represents the quantized latent feature, and $x_i'$ represents the pixel values of the reconstructed image.

## 4 EXPERIMENTAL RESULTS

### 4.1 Training details

The training details and the hyperparameter settings are described in this subsection. Our hardware platform is Intel Xeon E5-2690 V3 @2.60GHz CPU with 64GB memory and two Titan X GPUs. The method is implemented on the CompressAI platform [4]. The $\lambda$ belongs to $\{0.0018, 0.0035, 0.0067, 0.0130, 0.025\}$. For the training dataset, we utilize the JPEGAI dataset and 300,000 images randomly selected from the OpenImages dataset. Images are randomly cropped to a size of 256×256, with a batch size of 8. For entropy coding, we utilize the Range Asymmetric Numeral System [13]. The training process comprises two stages. In the pre-training stage,

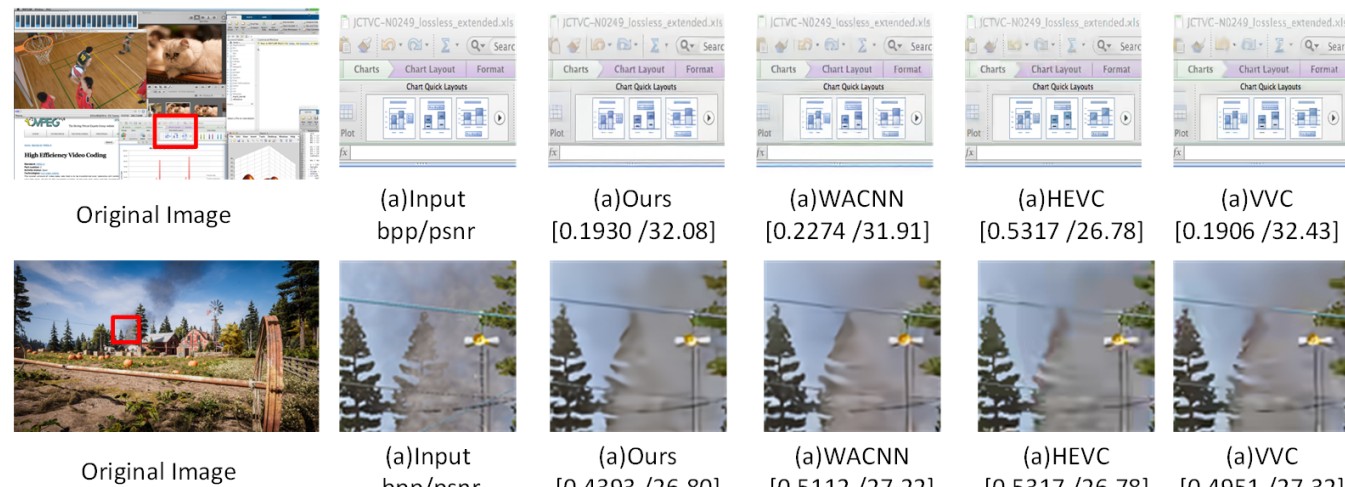

| | | | | | |
|---|---|---|---|---|---|
| Original Image | (a)Input bpp/psnr | (a)Ours [0.1930 /32.08] | (a)WACNN [0.2274 /31.91] | (a)HEVC [0.5317 /26.78] | (a)VVC [0.1906 /32.43] |
| Original Image | (a)Input bpp/psnr | (a)Ours [0.4393 /26.80] | (a)WACNN [0.5112 /27.22] | (a)HEVC [0.5317 /26.78] | (a)VVC [0.4951 /27.32] |

Figure 8: Qualitative results. Our method reconstructs the text regions, graphics, and wires with high fidelity, respectively.

Table 1: Comparison with existing compression methods on BD-Rate (%). The BD-Rate of BPG, VVC, cheng2020 [10], ms2020-2-Decoder [19], Dec-Adapter [29], WACNN [42] and our method are provided for comparison. The average performance is calculated by the utilized datasets, and tabulated in the last column. A smaller valuer is more effective.

| Method | JPEGAI | SIQAD | SCID | MLSCID | Average |
|---|---|---|---|---|---|
| BPG | +14.3 | +49.0 | +81.8 | +47.1 | +48.1 |
| VVC | -10.3 | -1.0 | +19.9 | +11.2 | +5.0 |
| cheng2020 [10] | +4.5 | +30.3 | +43.0 | +40.4 | +29.6 |
| ms2020-2-Decoder [19] | — | — | — | -4.1 | -4.1 |
| Dec-Adapter [29] | — | +2.9 | +92.4 | - | +47.7 |
| WACNN [42] | 0.0 | 0.0 | 0.0 | 0.0 | 0.0 |
| Ours | -6.3 | -14.7 | -15.7 | -16.3 | -13.3 |

300,000 images randomly chosen from OpenImages are used as the training set. The Adam optimizer with a learning rate set to $1 \times 10^{-4}$ for $120k$ iterations in this stage. The pre-training stage will stop until the learning rate drops to $2.7 \times 10^{-6}$. In the training stage, the pre-trained models are trained on the JPEGAI dataset. The optimizer and learning rate scheduling in the second stage are consistent with those in the pre-training stage.

## 4.2 Rate-distortion performance

We compute the peak signal-to-noise ratio (PSNR) and bits per pixel (BPP) for each image in the JPEGAI, SIQAD, and SCID datasets, and average the values to plot the rate-distortion curves as shown in Fig. 5, Fig. 6, and Fig. 7. The coding performance is measured by BD-Rate and negative values represent performance gains. We retrain the WACNN with the same training strategy. The images in JPEGAI are cropped into sizes less than 1920×1080 for inference. For Dec-Adapter [29] and Zou [41], we use the results from the Dec-Adapter [29] for comparison directly. For cheng2020 [10], we retrain it with the same training strategy as our method. For ms2020-2-Decoder [19], we evaluate the performance of the publicly available model. The proposed method is significantly better

than the baseline method. The WACNN results are obtained by our retrained models. Our method achieves an improvement of nearly 0.3dB on PSNR compared to the baseline on the JPEGAI dataset. Additionally, we perform a comparison with the learned methods and other conventional codecs. The results are presented in Table 1. We assess the performance of VVC under the all intra (AI) configuration. The version is VTM-10.0. It shows that our method achieves 13.3% BD-Rate saving on average.

The snapshots of reconstructed images using different compression methods at similar bitrates are shown in Fig. 8. Under approximately the same compression rate, the algorithm proposed in this paper preserves finer details and clearer structures over existing methods. It can be observed that our method restores more realistic character edges and thinner edges, demonstrating higher visual quality.

## 4.3 Ablation study

We perform the ablation study on the effectiveness of different methods proposed in Section 3. Our results are evaluated on the SIQAD dataset.

 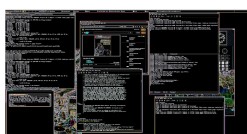 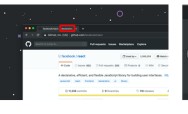

(a)binary ROI mask   (b)task-specific ROI mask   (c)structural prior

**Figure 9: The comparison of different priors. In the earlier SFT-based learned image compression [8], the external prior is the binary ROI mask shown in (a). Song et al. [30] employ the task-specific ROI mask with values in the range $[0, 1]$ as external priors shown in (b). We extract external prior from the gradient and text position information shown in (c).**

**Table 2: Comparison with SEM and SFT & CB on BD-Rate (%).**

| Method | BD-Rate |
| --- | --- |
| SFT & CB [30] | - |
| SEM | -24.7 |

**Effectiveness of SEM.** As shown in Fig. 9, it is evident that the external prior in our work is more abundant and complex than the previous prior. Hence, improvements of SFT & CB [30] are necessary to adapt to the difference. To show the effectiveness of SEM, we compare the SEM with SFT & CB. The results are tabulated in Table 2. The SFT & CB fails to generate efficient structural features leading to the quality loss in the table and the appearance of artifacts in Fig 3. In contrast, the SEM can handle complex external priors well, providing parameters for feature transform and leading to better performance.

**Table 3: Comparison with and without structural prior on BD-Rate (%).**

| Method | BD-Rate |
| --- | --- |
| W/O structural prior | - |
| W structural prior | -7.3 |

**Effectiveness of structural prior.** To evaluate the performance of EPG, we use uniform quality maps (W/O structural prior) [30] for comparison. The results are tabulated in Table 3. It brings improvement in compression quality for SCIs due to the enhancement of latent features by generating structural prior.

**Table 4: Comparison with and without text position prior on BD-Rate (%).**

| Method | BD-Rate |
| --- | --- |
| W/O text position prior | - |
| W text position prior | -5.7 |

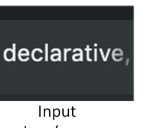 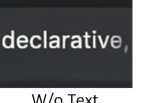

Original Image   Input bpp/psnr   W/o Text Position Prior [0.0802/38.32]   W Text Position Prior [0.0796/38.05]

**Figure 10: The ablation of text position prior.**

**Table 5: Comparison with different text detectors on BD-Rate(%).**

| Method | EAST | RDD | CRAFT |
| --- | --- | --- | --- |
| BD-Rate(%) | - | -0.41 | -1.2 |

**Table 6: Comparison WMSE based on text region mask with MSE on BD-Rate (ASIQE) (%).**

| Method | BD-Rate (ASIQE) |
| --- | --- |
| MSE | - |
| Our WMSE | -21.3 |

**Effectiveness of text position prior.** We compare our model with the W/O text position prior case. Table 6 shows the comparison results and Fig. 10 presents qualitative comparison results. As shown in Fig. 10, the text position prior leads to better integrity preservation in character "e". These results indicate that the text region mask effectively preserves text integrity and improves text region compression performance.

**Comparison of different text detectors.** The text region mask is decided by the text detector. Different text detectors are evaluated by the compression performance including EAST [40], CRAFT [1], and RRD [21]. As shown in Table 5, CRAFT shows the best performance. This is because of the precise text region mask provided by CRAFT. The CRAFT obtains 1.2% BD-Rate saving compared with EAST.

**Effectiveness of loss metric.** To verify the effectiveness of the proposed loss metric, we compare our WMSE loss function with MSE. Since ASIQE [16] is consistent with the visual quality of the image text context [32], we measure objective quality with ASIQE in this experiment. The compression performance on the SIQAD dataset is tabulated in Table 6. The model optimized with the proposed loss metric brings 21.3% BD-Rate (ASIQE) saving.

## 5 CONCLUSION

In this work, we propose a learned compression framework for SCIs. Our motivation is from the different characteristics between SCIs and natural images. To help preserve the structural fidelity and text integrity, the EPG is proposed which aims to extract global spatial attention for better SCI compression. Then, the SEM is proposed, which includes FEM and NESFT. It shows more reasonable bit allocation and a better visual effect than the SFT & CB method. Finally, a text region mask-guided rate distortion loss function is proposed to

give priority to the compression quality of text regions. Experimental results show that the proposed methods achieve 13.3% BD-Rate saving compared to the baseline WACNN on the JPEGAI, SIQAD, SCID, and MLSCID datasets on average. In our future work, we will investigate variable bit-rate learned SCI compression models and investigate hardware-friendly algorithms for real-time applications.

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
