# OpenReview forum: "Enhanced Screen Content Image Compression: A Synergistic Approach for Structural Fidelity and Text Integrity Preservation"
_acmmm.org/ACMMM/2024/Conference — MM2024 Oral_

### Official Review · Reviewer_f27U · 2024-05-08

**Rating:** 4
**Confidence:** 3

**Summary:**

The paper explores a global structural and textural prior based screen content image compression method. The proposed method consists of two part, an external prior guidance module and a structural enhancement module. Based on promising design of the network, the proposed method can get 13.3% BD-rate gain over WACNN in CVPR’22.

Novelty lies in several aspects. Firstly, the method proposed an external prior guidance module, which use laplace gradient operator to detect the region of text and use the text region as prior to fruther refine the basic structual prior and guide the distinguishment of text region and non-text region. Secondly, the paper proposed a structural enhancement module which consists of structural enhancement part and transformation part, with structural enhancement part further highlighting the textual part in image and using transformation part to extract these high frequency textual part to improve the total coding performance.

**Strengths:**

The paper is well-written and well-organized. How to efficiently predict and reconstruct the text part with strong structure and texture has always been an important part of screen content image compression. This paper proposes a module to distinguish between text and non-text regions and a module to extract structural and texture information, and improve the compression performance by emphasizing the compression of such information. In addition to this, it gives a clear theoretical foundation as well as a presentation of the relevant details thus supporting the validity of the method proposed in this paper.

**Limitations:**

First, there are some formatting problems with this article, such as “ACM reference format...” should be removed from the text. Secondly, I'm curious about the performance of this approach compared to enhanced compression model (ECM) with the screen content encoding tool fully enabled, as ECM makes a number of interesting enhancements to the screen content tools compared to VVC. It would be more beneficial to support the proposed method if some comparison of newer and better screen content compression methods could be given.

**Suitability:**

3

---

### Official Review · Reviewer_gzMx · 2024-05-16

**Rating:** 5
**Confidence:** 3

**Summary:**

This paper proposes an enhanced screen content image (SCI) compression approach, in which the structural fidelity and text integrity are preserved. Considering the unique characteristics of SCIs, this paper emphasizes optimizing the bit allocation for SCI compression and minimizing text region blurring problem. Experimental results show that the proposed method surpasses the performance of other baselines on BD-Rate saving.

**Strengths:**

1. The external prior guidance (EPG) module is proposed to enhance structural fidelity and text integrity by providing global spatial attention. The Laplace gradient operator and text detectors are employed to formulate the external priors.
2. A structural enhancement module (SEM) is introduced to improve the preservation of structural information by enhanced spatial feature transform. In SEM, the feature extraction module (FEM) and nonlinear enhancement spatial feature transform (NESFT) is integrated to distinguish the structural prior features and propagate them into the backbone.
3. The weighted mean square error (WMSE) loss function is proposed for better compression efficiency in text regions.

**Limitations:**

The SEM incurs additional computational overhead, and the paper does not consider the computational complexity in the experiments.
The appearance of names at the top of the pages suggests there may be an issue with the template configuration.

**Suitability:**

3

---

### Official Review · Reviewer_8SaL · 2024-05-24

**Rating:** 4
**Confidence:** 2

**Summary:**

1. An External Priori guidance (EPG) module is proposed to provide global spatial attention by combining basic structure priors and text position priors to enhance structure fidelity and text integrity.
2. Structure enhancement module (SEM) is proposed, including feature extraction module (FEM) and nonlinear enhanced spatial feature transform (NESFT). FEM extracts prior structure information, and NESFT fuses it into the backbone network to improve the retention of structure information.
3. In the loss function, the weighted mean square error (WMSE) is used instead of MSE to give a higher weight to the text area, which improves the compression efficiency of the text area.

**Strengths:**

1. This work focuses on the important but under-addressed area of image compression for screen content and proposes a novel approach to end-to-end structural fidelity and text integrity retention.
2. By introducing external prior guidance and structure enhancement module, the existing method effectively solves the problem of structural information and text detail loss when processing screen content images, and significantly improves the compression performance.
3. EPG module uses basic structure and text position prior to provide global spatial attention; The SEM module is designed with reasonable feature extraction and nonlinear enhancement to realize effective use of prior structure information.

**Limitations:**

1. The paper lacks specific implementation details of some key modules, such as the specific network structure and parameter Settings of FEM and NESFT, which may affect the repeatability of the work.
2. The paper does not analyze the computational complexity and model size of the proposed method, which are crucial for practical applications.

**Suitability:**

2

---

### Meta-Review · Area_Chair_hYRi · 2024-07-02

**Recommendation:** Accept (Oral)
**Confidence:** 4

**Metareview:**

This paper proposed a novel image compression approach for screen content.  All reviewers appreciate the novelty of the work, but share the concerns around complexity and performance of the proposed approach.  Implementation details and more performance analysis would be helpful.